# Pathophysiology and Treatment Options for Hepatic Fibrosis: Can It Be Completely Cured?

**DOI:** 10.3390/cells10051097

**Published:** 2021-05-04

**Authors:** Arshi Khanam, Paul G. Saleeb, Shyam Kottilil

**Affiliations:** 1Division of Clinical Care and Research, Institute of Human Virology, University of Maryland School of Medicine, Baltimore, MD 21201, USA; akhanam@ihv.umaryland.edu; 2Institute of Human Virology, University of Maryland School of Medicine, Baltimore, MD 21201, USA; PSaleeb@ihv.umaryland.edu

**Keywords:** liver fibrosis, Kupffer cells, hepatocytes, hepatic stellate cells, inflammation, extracellular matrix, inflammasomes, exosomes, miRNA

## Abstract

Hepatic fibrosis is a dynamic process that occurs as a wound healing response against liver injury. During fibrosis, crosstalk between parenchymal and non-parenchymal cells, activation of different immune cells and signaling pathways, as well as a release of several inflammatory mediators take place, resulting in inflammation. Excessive inflammation drives hepatic stellate cell (HSC) activation, which then encounters various morphological and functional changes before transforming into proliferative and extracellular matrix (ECM)-producing myofibroblasts. Finally, enormous ECM accumulation interferes with hepatic function and leads to liver failure. To overcome this condition, several therapeutic approaches have been developed to inhibit inflammatory responses, HSC proliferation and activation. Preclinical studies also suggest several targets for the development of anti-fibrotic therapies; however, very few advanced to clinical trials. The pathophysiology of hepatic fibrosis is extremely complex and requires comprehensive understanding to identify effective therapeutic targets; therefore, in this review, we focus on the various cellular and molecular mechanisms associated with the pathophysiology of hepatic fibrosis and discuss potential strategies to control or reverse the fibrosis.

## 1. Introduction

Liver fibrosis is associated with significant morbidity and mortality and is an important contributor to the global disease burden [1]. Fibrosis is characterized by the accumulation of collagen and other extracellular matrix (ECM) components that are indispensable for wound healing and tissue repair in various conditions, including injury, infection, inflammation, autoimmune disease and tumors [2,3,4,5,6,7]. However, fibrotic remodeling may impair organ function and encourage further disease progression [8]. The pathogenesis of hepatic fibrosis is multifaceted and accompanies progressive liver injury that varies from mild to severe. It arises as a consequence of diverse biologic processes induced by alcoholic liver disease (ALD), non-alcoholic fatty liver disease (NAFLD), non-alcoholic steatohepatitis (NASH), autoimmune hepatitis (AIH), drug-induced liver injury (DILI) and viral hepatitis (hepatitis B and hepatitis C) [9,10,11,12,13,14,15,16]. Increasing rates of obesity have further accelerated the risk of liver fibrosis due to liver injury caused by NAFLD and NASH [17,18].

Hepatic fibrosis epitomizes a universal response of the liver to acute or chronic liver injury. After acute liver damage, parenchymal cells replace necrotic and apoptotic cells by the process of regeneration; however, during chronic liver injury, the regenerative response fails gradually, and ECM leading to hepatic fibrosis increasingly substitutes hepatocytes [19]. During the progression of chronic liver injury towards fibrosis, most hepatic cells, including parenchymal as well as non-parenchymal cells, undergo specific changes [20]. The injured hepatocytes undergo apoptosis, while sinusoidal endothelial cells experience the loss of fenestrae, known as capillarization of the sinusoids [21]. Kupffer cells (KCs) and resident liver macrophages are activated, producing a wide range of cytokines and chemokines [22,23]. Lastly, quiescent hepatic stellate cells (HSCs) are also activated, and express new receptors and proteins, such as platelet derived growth factor (PDGF) receptor, transforming growth factor-β (TGF-β) receptor, and ex α-smooth muscle actin (α-SMA). The activated HSCs proliferate and produce ECM proteins to form a fibrous scar [24,25,26].

The hepatocyte response to inflammation plays a pivotal role in the pathophysiology of hepatic fibrosis. Following liver injury, infiltration of monocytes, macrophages, neutrophils, natural killer (NK) cells, CD4 and CD8 T cells occurs [27,28]. These cells are activated by pathogen associated molecular patterns (PAMPs) or danger associated molecular patterns (DAMPs), released by injured or damaged cells and produce a range of pro-inflammatory mediators leading to hepatic inflammation [29]. Subsequently, these inflammatory mediators activate HSCs, which then differentiate into myofibroblasts and become the primary source of ECM [30]. HSC participates in the inflammation process by interacting with various immune cells. In case of chronic hepatitis B (CHB) infection, the host immune response induces extensive hepatocyte damage, leading to fibrosis [31]. Recently, it has been demonstrated that inflammasomes accelerate hepatic inflammation [32]. Inflammasomes are intracellular multi-protein complexes that are expressed in both parenchymal as well as non-parenchymal cells including hepatocytes and are crucial regulators of inflammation. These inflammasomes respond to danger signals, release pro-inflammatory cytokines IL-1β and IL-18, and initiate a unique pathway of programmed cell death, termed as pyroptosis [33]. Nucleotide-binding domain leucine-rich repeat (NLR) family pyrin domain containing 3 (NLRP3) inflammasome plays an important role in the development of liver fibrosis by directly activating HSCs [34,35]. Uncontrolled fibrosis is associated with further disease progression to cirrhosis and hepatocellular carcinoma (HCC); therefore, control and mitigation of fibrosis is critical [36]. There is no standard treatment for liver fibrosis; hence, there is a critical need to explore therapies that reduce liver fibrosis and stimulate liver regeneration.

This review provides an update of the current knowledge of cellular and molecular mechanisms involved in the pathophysiology of hepatic fibrosis and explores potential therapeutic approaches for its mitigation.

## 2. Triggers of Hepatic Fibrosis

The development of liver fibrosis is navigated by ongoing liver injury that occurs through several mechanisms. Chronic liver injury due to viral hepatitis, ALD, NAFLD, NASH, AIH and drug-induced hepatotoxicity provoke liver fibrosis, where hepatitis B, C and alcohol abuse remains the most common cause. Depending on the underlying etiology, liver fibrosis displays different morphological patterns. For instance, viral hepatitis is associated with interface hepatitis and portal-central vein bridging fibrosis [37], while alcoholic fibrosis and NASH are characterized by perisinusoidal or pericellular fibrosis [38]. However, few mechanisms are common across etiologies. After liver injury, HSC activation is a common process; HSCs are activated and differentiate into myofibroblasts like cells. Activated KCs, monocytes, platelets and damaged hepatocytes secrete PDGF and TGF-β; initiating intracellular signaling cascades leading to HSC activation. While the key pathways of HSC activation are common to all forms of liver injury and fibrosis, Disease-specific pathways also exist. Pathophysiological mechanisms that are discrete among different etiologies emphasize the need for a separate therapeutic approach for liver fibrosis.

### 2.1. Viral Hepatitis

In chronic viral hepatitis, fibrosis progression is a dynamic process. HSC, the most important contributor of hepatic fibrosis, responds to different viral proteins and host factors. It is a well-known fact that both HBV and HCV are non-cytopathic viruses and liver injury is mainly driven through host immune response. Persistent viral exposure activates different immune cells, producing numerous pro-inflammatory cytokines and fibrogenic mediators, and recruiting other immune cells to the liver driving hepatic inflammation [39]. Viral proteins as well as pro-fibrogenic mediators stimulate HSC activation and subsequently induce fibrosis development. Moreover, continuous hepatic inflammation induces ECM deposition. One study reported that a combination of serum HBV DNA, alanine aminotransferase level and gender may effectively identify the patients who are at higher risk of developing fibrosis [37]. Another set of data revealed HBV DNA (cut off value 4.91 log IU/mL) as a predictor of significant fibrosis in HBeAg negative CHB patients [38]. Moreover, few HCV proteins are capable of directly inducing fibrosis by the activation of pro-inflammatory and pro-fibrogenic properties of HSCs [40]. HCV proteins stimulate the production of reactive oxygen species (ROS) that contribute to fibrogenesis by causing direct damage to hepatocytes as well as through the activation of HSCs. In HCV, clearance of activated HSCs through apoptosis is considered a key determinant of fibrosis reversion.

### 2.2. Alcoholic Liver Disease (ALD)

Long histories of excessive alcohol consumption causes ALD. In ALD, toxic ethanol metabolites such as acetaldehyde mediate hepatic fibrosis through hepatocyte apoptosis and ROS production. In the hepatic cytosol, alcohol is metabolized into acetaldehyde by alcohol dehydrogenase, which then converts to acetate by acetaldehyde dehydrogenase. Acetaldehyde is considered the main toxin in alcohol-mediated liver injury, including inflammation, cellular damage, ECM remodeling and fibrogenesis [39]. In addition, acetaldehyde triggers TGF-β1 dependent late phase response in HSCs, which maintains a pro-fibrogenic and pro-inflammatory state [40]. Acetaldehyde can directly activate HSCs and encourage collagen I expression [41]. Furthermore, extreme alcohol abuse increases the gut permeability allowing the translocation of bacteria derived lipopolysaccharide (LPS) from the gut to the liver; LPS then stimulates HSC activation after binding to TLR4 and stimulating KCs [42]. These cells produce cytokines for HSC activation causing fibrosis. Moreover, LPS binding to TLR4 activates the MyD88-independent toll-interleukin-1 receptor domain-containing adaptor-inducing interferon-β/IRF-3 signaling pathway, which yields oxidative stress and pro-inflammatory cytokines causing hepatocellular damage and contributes to alcoholic steatohepatitis [41,42]. Previous data reported that TLR4 signaling activation in HSCs and LSECs promote fibrogenesis [43]. Activation of TLR4 signaling in LSECs regulate angiogenesis through MyD88-effector protein, which regulates extracellular protease production, and that in turn results in fibrosis [44].

### 2.3. NAFLD/NASH

NAFLD is characterized by the excess fat accumulation in the liver of individuals who consume little or no alcohol, while NASH is a progressive form of NAFLD which occurs in the context of the metabolic syndrome, and is characterized by hepatic steatosis leading to the development of fibrosis [45]. Overflow of free fatty acids in the liver induce ROS production and subsequently oxidative stress that acts as a critical factor in driving fibrosis by numerous pathways. Oxidative stress impedes replication of mature hepatocytes leading to the accumulation of immature progenitor cells, resulting in the formation of small ductules [46]. This ductular reaction is associated with the development of fibrosis in NAFLD/NASH through the secretion of pro-inflammatory cytokines and epithelial to mesenchymal transition of cholangiocytes to fibrogenic myofibroblasts [47,48].

### 2.4. Autoimmune Liver Disease

Autoimmune liver diseases occur when the immune system loses self-tolerance and attacks the liver. Loss of self-tolerance encourages autoimmune cell damage and hepatic inflammation. The three main categories of autoimmune liver disease include autoimmune hepatitis, primary biliary cirrhosis (PBC) and primary sclerosing cholangitis (PSC) [49]. During PBC and PSC, the accumulation of bile acids mediates fibrogenesis through the induction of cholangiocytes, hepatocytes apoptosis and necrosis, and activation of bile acid [50]. Depletion of FXR inhibits cholestatic fibrogenesis, suggesting FXR as a critical therapeutic target [50]. Moreover, cholangiocytes produce a wide variety of pro-fibrogenic factors that act on portal myofibroblasts leading to hepatic fibrosis [51].

## 3. Role of HSCs in Liver Fibrosis

HSCs, also known as perisinusoidal cells, are the central effectors and play a crucial role in hepatic fibrosis. These cells are found in the perisinusoidal space of the liver, the area between hepatocytes and sinusoidal endothelial cells, and are closely associated with both of these cell types. Under normal liver conditions, HSCs encompasses approximately 15% of the total number of liver-resident cells and one third of the non-parenchymal cell population. Generally, HSCs serve as the major storage site for vitamin A; however, they also respond to hepatic injury and play a crucial role in hepatic fibrosis that ruins several aspects of hepatic function.

Usually, HSCs are quiescent; however, after liver damage, hepatocytes and immune cells release different components leading to the activation of HSCs, which is an essential and central step of liver fibrogenesis [7]. Activated HSCs act as the main source of active myofibroblasts that drive the fibrogenic process [52]. Many chronic liver injuries, including viral hepatitis, ALD, AIH, toxins, as well as NAFLD/NASH can trigger HSC activation into myofibroblasts like phenotype. HSCs are activated to become proliferative and contractile myofibroblasts. They are involved in the initiation, progression and degeneration of liver fibrosis by secreting different fibrogenic factors. Activated HSCs secrete extracellular matrix proteins, including collagens type I, III and IV, proteoglycans, glycoproteins, fibronectin and laminin and facilitate healing of the damaged liver. Nevertheless, continuous HSCs activation produces abundant collagen, leading to hepatic fibrosis and impaired hepatic function. Several factors, including inflammatory stimuli, fibrogenic cytokines TGF-β, PDGF, epidermal growth factor (EGF), ROS, produced by activate macrophages, KCs, platelets and products of damaged hepatocytes drive HSC activation [53,54]. After hepatic damage, hepatocyte necrosis as well as apoptosis occurs, which further promotes HSC activation[55]. Evidence suggests that apoptotic fragments released from hepatocytes have fibrogenic potential towards cultured stellate cells, while lipid peroxidases produced from necrotic hepatocytes also induce HSC activation [56,57]. Moreover, activated HSCs are also involved in the chemotaxis process by producing chemoattractant that further aggravates hepatic injury. The ultimate outcome of all these processes is increased ECM accumulation and replacement of the normal matrix [58]. Cytokines released by HSCs can augment the inflammatory and fibrogenic tissue responses while matrix proteases may accelerate the replacement of normal matrix [59].

HSCs induce fibrosis, not only by increasing cell number, but also by increasing ECM production per cell. In healthy livers, the matrix primarily contains collagen IV and VI, which is progressively replaced by collagens I and III as well as fibronectin during fibrogenesis [1]. In addition, activated HSCs secrete connective tissue growth factor (CTGF) that gives major fibrogenic signals. CTGF substantially correlates with liver fibrosis in HCV patients and can be served as a valuable serum biomarker for fibrosis severity [60,61]. Activated HSCs are also characterized by enhanced survival that is mediated by KCs in a nuclear factor-kappa B (NF-kB) dependent manner that further promotes fibrosis [62]. Nonetheless, inhibition of NF-kB pathways also inhibits hepatic fibrosis by inducing HSC apoptosis; therefore, stimulation of HSC apoptosis could be a potential approach for the treatment of hepatic fibrosis [63]. TGF-β has long been considered as one of the most powerful cytokines to induce fibrosis [64]. Hepatocytes, macrophages, liver sinusoidal endothelial cells (LSEC) and other hepatic regulatory cells produce TGF-β. Normally hepatic cells produce a lower amount of TGF-β; however after chronic liver injury, secretion of TGF-β increases tremendously. HSCs are activated after receiving signals from TGF-β; simultaneously they also secrete TGF-β that acts through an autocrine positive feedback mechanism. Binding of TGF-β to its receptor leads to the phosphorylation of SMAD proteins including SMAD 2 and 3, which binds to SMAD 4 before translocating to the nucleus. This leads to the induction of myofibroblasts and matrix deposition [65]. TGF-β induces quiescent HSCs trans-differentiation into myofibroblasts that secrete ECM. Additionally, TGF-β activates several other signaling pathways, including mitogen activated protein kinase (MAPK), phosphatidylinositol-3- kinase/AKT and rho GTPase pathways where MAPK pathway is potentially involved in hepatic fibrosis. While hepatocytes apoptosis and necrosis encourage HSC fibrogenesis, TGF-β endures hepatocyte mass and regulates growth during the process of liver regeneration. Hence, blocking TGF-β might be challenging, as few of its functions, such as its anti-inflammatory nature and growth regulatory function, are critical to maintain liver homeostasis. 

## 4. Role of LSECs in Hepatic Fibrosis

LSECs are highly specialized endothelial cells that contain fenestrations and play a critical role in maintaining liver homeostasis. LSECs line the wall of hepatic sinusoid and are persistently exposed to antigens carried from gastrointestinal tract through the portal vein. Function of LSEC includes removal of macromolecules and small particulates from the blood. They are also involved in liver regeneration, hepatic fibrosis and interactions with tumor metastasis [66]. These cells possess immunological roles and exhibit high endocytic activity, endocytosing cellular and extracellular components rapidly by clathrin-mediated endocytosis. Hepatic damage drives molecular changes leading to phenotypic alterations and abnormalities in LSECs. Impaired LSEC function has been reported in chronic liver disease (CLD) [67]. Dysregulated LSECs response leads to chronic inflammation and further drive hepatic fibrosis. Moreover, these cells induce phenotypic changes in HSCs and are probably one of the initial triggers of HSC activation that alter ECM to increase tissue stiffness, which in turn activates both LSECs and HSCs. Moreover, LSECs are also involved in fibrosis resolution by maintaining HSCs and KCs in a quiescent state. 

Before the initiation of fibrosis, a change in LSECs takes place that is termed as capillarization or dedifferentiation, in which LECs lose fenestration. Capillarization induces hepatic fibrosis in ALD in humans [68], NAFLD/NASH in mice [69]. As capillarization of LSECs occurs prior to fibrosis, their function to regulate HSC activation is lost. A study reported that co-culture of healthy LSECs with HSCs prevents HSC activation and reverses activated HSCs into quiescent state. While the co-culture of capillarized LSECs with HSCs did not prevent HSC activation, suggesting LSECs as a critical regulator of HSC activation [70]. Vascular endothelial growth factor (VEGF) pathways, including endogenous nitric oxide synthase (eNOS)-soluble guanylate cyclase (sGC)-cGMP pathway and an undefined VEGF-dependent, nitric oxide (NO)-independent pathway, preserve fenestration of LSECs. During liver fibrogenesis, occurrence of angiogenesis has been reported. Liver angiogenesis exacerbates hepatic fibrosis, and in turn, liver fibrosis augments angiogenesis [71]. Most anti-angiogenic agents possess anti-fibrotic potential. Moreover, LSECs release angiocrine signals that are critical for maintaining a balance between liver regeneration and fibrosis. During chronic liver injury, LSECs persistently express fibroblast growth factor receptor 1 (FGFR1), supporting CXCR4-driven fibrotic angiocrine responses and promote liver fibrosis. However, during acute liver injury, CXCR7-Id1 pathway activation induces the production of hepato-active angiocrine factors resulting in liver regeneration [72]. Hence, differentially stimulated LSECs establish different angiocrine signals to balance liver regeneration and fibrosis. Restoration of LSEC phenotype by inhibiting mechano-sensitive pathways could be fascinating for therapeutic approaches for the treatment and reversal of liver fibrosis.

## 5. Role of Liver-Resident Macrophages in Hepatic Fibrosis

Liver-resident macrophages, known as KCs, are located at the luminal side of the hepatic sinusoidal endothelium. They occupy fixed positions and recognize their microenvironment through long cytoplasmic extensions. KCs are the critical component of innate immune mononuclear phagocytic systems and perform crucial functions during homeostasis, including the clearance of systemic or gut-derived pathogens, and act as first responders following liver injury [73]. Since KCs exhibit plasticity and are critical mediators of liver injury and fibrosis, they have been proposed as potential therapeutic targets [74,75]. Under diverse pathological conditions, they differentiate into pro-inflammatory M1 (classical) and anti-inflammatory M2 (alternative) KCs. M1 KCs are typically induced by IL-12, IFN-γ and LPS produce IL-1β, IL-6, IL-12 and TNF-α, and are closely associated with Th1 activation. M2 KCs are controlled by IL-4, IL-10 and IL-13, secrete IL-10, TGF-β, PDGF and EGF, and are linked to Th2 priming. The balance of M1 and M2 KCs regulate hepatic inflammation [75,76]. Under pathophysiological conditions, M2 KCs create an anti-inflammatory milieu and promote tissue repair through ECM remodeling and recruitment of fibroblasts [77,78]. During this process, the balance between TGF-β-dependent deposition of new ECM and MMP-mediated degradation promotes tissue repair but not fibrosis. However, when the lesion persists, M2 KCs acquire a pro-fibrotic role by secreting a large amount of TGF-β and galectin-3, inducing fibrosis [79].

During the process of inflammation, KCs expand and secrete IL-1β, TNF-α, CCL2 and CCL5, ensuing paracrine activation of either protective or apoptotic signaling pathways and allowing the recruitment of other immune cells that may aggravate hepatic injury [80]. Along with the inflammatory stimuli, metabolic signals also modulate hepatic KCs activation [81]. The location of KCs in the sinusoids allows close interactions with other non-parenchymal hepatic cell populations. In the context of hepatic fibrosis, KCs employ a dual function by either promoting or retracting the excessive deposition of ECM [82,83]. Several studies have shown that KCs activate HSCs through a paracrine mechanism involving the potent pro-fibrotic and mitogenic cytokines TGF-β and PDGF to transdifferentiate them into myofibroblasts, which are the main collagen-producing cell type in hepatic fibrosis [84]. Furthermore, KCs can express numerous matrix metalloproteinases (MMPs) containing MMP-9, MMP-12 and MMP-13 that are involved in matrix degradation, resulting in resolution of liver injury and fibrosis [85]. The phagocytic and endocytic ability of KCs make them an easy target for biofunctionalized nanoparticles intended to impact macrophage polarization and serve as carrier tools for drug delivery [86]. However, to translate such concepts into clinical application, specific contributions of KCs to liver injury and fibrosis need to be fully understood. During viral hepatitis, KCs can provide an efficient antiviral response but can also contribute to adverse effects by mediating hepatic fibrosis and the suppression of antiviral immunity [87]. Margatoxin is a 39 amino acid peptide, which selectively inhibits voltage-activated potassium channels (Kv.13). Recently, it has been revealed that margatoxin protects mice from CCL4-induced liver fibrosis by decreasing the expression of M1 macrophage phenotype and increasing M2 macrophage. In addition, margatoxin inhibits the production of macrophage pro-inflammatory cytokines by suppressing p-STAT1 activity and promoting IL-10 secretion by increasing p-STAT-6 activity. The study concludes that margatoxin alleviates CCL4 induced hepatic fibrosis in mice, possibly through macrophage polarization, cytokine secretion and STAT signaling [88]. KCs express T Cell Immunoglobulin and Mucin-4 (TIM-4), which is associated with the progression of liver fibrosis [89]. Inhibition of TIM-4 with mAB or TIM-4 siRNA decreases liver fibrosis by lessening hydroxyproline and collagen deposition in CCL4-induced liver fibrosis. Moreover, disruption of TIM-4 signaling inhibits macrophage migration/function and suppresses TLR2/4/9 dependent signaling, which are important for regulating the immune function of macrophages [90]. TIM-4 mediated Akt1/mitophagy in KCs is linked to pro-fibrotic polarization, apoptotic resistance and development of fibrosis. TIM-4 interference in KCs provides a novel immune target to prevent liver fibrosis progression. Exosomes originating from LPS treated THP-1 macrophages promote HSC proliferation and induce fibrosis by encouraging the expression of fibrotic genes. LPS mediated THP-1 macrophage activation changes the miRNA profile of exosomes. miR-103-3p activation in exosomes promotes HSC activation and proliferation by targeting Kruppel-like factor 4 (KLF4) and plays an important role in the crosstalk between THP-1 macrophages and HSCs during the progression of liver fibrosis [91]. Macrophages not only activate HSCs but also support their survival. Interferon regulatory factor 5 (IRF5) has emerged as an important pro-inflammatory transcription factor involved in macrophage-mediated liver fibrosis and is a marker of liver damage [92] Mice lacking IRF5 are protected from hepatic fibrosis induced by metabolic or toxic stress. Moreover, transcriptional reprogramming of macrophages lacking IRF5 results in immunosuppressive and anti-apoptotic properties. IRF knockout mice respond to hepatocellular stress by promoting hepatocyte survival and providing complete protection from hepatic fibrogenesis. Since IRF5 mediates hepatocyte death and liver fibrosis in mice as well as humans, modulation of IRF5 function may be another attractive approach to treat fibrosis. Finally, involvement of the Janus Kinase (JAK)-2 pathway in the pathogenesis of hepatic fibrosis, steatosis, ischemia-perfusion injury and HCC has been observed [93]. In the liver, the JAK2 pathway plays a critical role in regulation of multiple processes including cell growth, differentiation, proliferation and immune functions by activating growth hormones and cytokines including IFN-γ, IL-4, IL-6, IL-12, IL-13 and leptin [94]. Selective JAK2 antagonist TG1011348 inhibits macrophage infiltration, expression of inflammatory markers and nitric oxide release from macrophages and attenuated HSC activation and collagen accumulation, suggesting its use as a potential therapy [93].

## 6. Role of Liver-Resident Lymphocytes in Hepatic Fibrosis

Studies have reported the presence of several types of lymphocytes in the liver. Unconventional T cells such as mucosal associated invariant (MAIT) cells, invariant natural killer T (NKT) cells, hepatic innate lymphoid cells (ILCs), γδ T cells, and memory CD8 T cells reside in different hepatic tissues [95]. Although different types of tissue resident lymphocytes share similar characteristics, the distinctive intrahepatic microenvironment can remodel their phenotypic and functional properties. Liver-resident lymphocytes’ functions include immunosurveillance and maintenance of liver homeostasis. However, under pathological conditions, liver-resident lymphocytes employ both protective as well as pathogenic effects and may contribute to hepatitis, fibrosis and cirrhosis.

NKT cells are enriched among liver lymphocytes and constitute 5–10% of human liver lymphocytes. NKT cells directly kill the target cells, produce a wide variety of cytokines, and play diverse roles in liver injury, fibrosis, regeneration and hepatocarcinogenesis. There are many types of NKT cells, including type I and type II. Generally, Type I NKT cells have a pro-inflammatory function in chronic liver disease, recruiting neutrophils and myeloid cells and encouraging the activation of HSCs, resulting in hepatocyte necrosis, fibrosis and even HCC [96]. Moreover, NKT cells produce large amounts of pro-fibrotic cytokines including IL-4 and IL-13 [96]. Type I NKT cells also inhibit liver regeneration by producing high levels of IFN-γ. Conversely, type II NKT cells suppress the pro-inflammatory response prompted by type I NKT and protect against liver damage [97]. NKT cells produce both pro-fibrotic and anti-fibrotic cytokines in HCV patients who do not have cirrhosis, whereas in those with cirrhosis, they produce only pro-fibrotic cytokines, suggesting that NKT cells promote fibrosis in HCV patients [98].

Hepatic innate lymphoid cells (ILCs) also display a pro-fibrotic role during chronic liver disease. ILCs are a family of innate immune lymphocytes that possess the phenotypes and function of T cells, and are involved in lymphoid tissue development, liver repair and homeostasis. Conventionally, ILCs have been classified into three subsets based on their development and function, one of which is ILC2, that have a pro-fibrotic effect [99]. ILC2s mainly produce Th2 associated cytokines, including IL-4, IL-5, IL-9 and IL-13 where IL-4 and IL-13 are the most fibrogenic. These cells also produce amphiregulin, which promotes tissue repair but may also contribute to ongoing fibrosis. During chronic liver injury, intrahepatic activation of ILC2s occurs via a ST2-dependent signaling pathway [100]. Activated ILC2s secrete IL-13 that induces HSC activation, exacerbates hepatic fibrosis, and further worsens liver function [101]. Nevertheless, other studies reveal a protective role of liver-resident ILC2s in tissue damage by mediating tissue healing and repair [102]. In an adenovirus-induced liver hepatitis model, ILC2s were amplified by increased expression of IL-33 and its receptor ST2, and reduced T cell mediated liver injury by attenuating TNF-α production [103]. Moreover, ILC2s prevent Th1 and Th17 derived inflammatory response and protect against con-A induced hepatitis [104]. Similarly, ILC3s exhibit a pro-fibrotic role in both mouse and CHB human liver, by producing Th17 cell associated cytokines including IL-17A, IL-17F and IL-22 [105]. ILC3s also endorse HSC mediated fibrogenesis by suppressing IFN-γ production by other immune cells and exerting indirect fibrogenic effects. In experimental liver fibrosis, IL-17A activates both HSCs and KCs, and stimulates TNF-α, IL-6 and TGF-β secretion that further aggravate fibrosis. IL-17A and IL-17RA deficient mice demonstrate reduced liver fibrosis, suggesting a pro-fibrotic role of IL-17 [106]. Conversely, IL-17A also has an anti-fibrotic role in fibroblasts by downregulating the expression of collagen and CTGF [107]. Therefore, targeting ILCs may represent a novel therapeutic strategy for the treatment of liver fibrosis.

γδ T cells are discrete subgroups of T cells expressing T cell receptors γ and δ chains with varied structural and functional heterogeneity. These cells are enriched in liver, represent 3–5% of total liver lymphocytes, and play an essential role in various liver diseases. γδ T cells have a protective role in liver fibrosis, as demonstrated by directly killing activated HSCs via NKp46 mediated release of cytolytic granules, and FasL mediated apoptosis during murine liver fibrosis. Deficiency of γδ T cells exacerbates hepatic fibrosis in the murine model [108]. Moreover, γδ T cells promote anti-fibrotic ability of conventional NK cells and liver-resident NK cells by improving their cytotoxic activity against activated HSCs. However, other studies highlighted the pathogenic role of γδ T cells through IL-17A mediated HSCs activation in CCL4 induced liver injury and fibrosis [109].

## 7. Role of Exosomes in Hepatic Fibrosis

Exosomes are small (~100 nm) membrane-bound extracellular vesicles originating from an endosomal pathway from various types of cells. They release from cells into extracellular space upon fusion with the plasma membrane and contain proteins, hormones, and various RNA species including miRNA as cargo molecules, delivering them to the target cells. Growing evidence indicates an increased release of exosomes in several inflammatory diseases [110]. Exosomes secreted in response to different inflammatory stimuli vary in their protein and miRNA content compared to the exosomes that are produced under homeostasis. In the liver, both parenchymal as well as non-parenchymal cells release exosomes. Hepatocytes release exosomes to the neighboring hepatocytes or non-parenchymal cells to regulate liver regeneration and repair. Exosomes released by hepatocytes fuse with target hepatocytes, and deliver sphingosine kinase 2 to form sphingosine-1- phosphate and promote cell survival and growth of the target hepatocytes [111]. Moreover, non-parenchymal cells, including HSCs, LSEC and cholangiocytes, secrete exosomes and regulate liver remodeling after injury [112]. In liver pathophysiology, crosstalk between similar or different cell types is an inevitable process through exosomes and exosomal proteins, mRNA and miRNA that modify the microenvironment in target cells. Hepatocytes also communicate with HSCs via exosomes. After liver injury, damaged hepatocytes release exosomes that are internalized into neighboring HSCs and give distress signals, leading to trans-differentiation of quiescent HSCs into activated myofibroblastic HSCs [113]. Furthermore, exosomes released from damaged hepatocytes carry unknown ligands that bind to TLR3 present on HSCs and activate them resulting in fibrosis. TLR3 activation on HSCs exacerbates liver fibrosis by augmenting the production of CCL-20 and IL-17A. Likewise, lipid induced cytotoxicity stimulates hepatocytes to release exosomes, retaining a fibrosis-inducing signal to HSCs [114]. Exosomal miR-192 significantly increases pro-fibrotic markers in HSCs. Exosomes from HCV-infected patients possess miR-19a, which targets the suppressor of cytokine, signaling 3 (SOCS3) to activate the signal transducer and activator of transcription-3 (STAT3) mediated TGF-β signaling pathways and activate HSCs. Exosomes may also inhibit progression of fibrosis. Exosomes derived from human mesenchymal stem cells (MSCs) alleviate CCL4 induced fibrosis in mice by inhibiting TGF-β and collagen expression. These exosomes also induce cellular senescence in cultured human HSCs, reducing the production of ECM components and inhibiting fibrosis. Quiescent HSCs release exosomes containing miR-214 with anti-fibrotic properties [115]. When these exosomes are transferred intracellularly in primary activated HSCs, they suppress CTGF, α-SMA and collagen, thereby reducing fibrosis. Moreover, quiescent HSC-derived miR-214-containing exosomes are taken up by hepatocytes, and downregulate CTGF expression in recipient hepatocytes. Conversely, exosomes derived from activated HSCs promote liver fibrosis, as they contain higher CTGF, mRNA and protein, suggesting that the components of exosomes determine their role in hepatic fibrosis [116]. LSECs also secrete exosomes: since LSECs are located near HSCs, exosomes derived from LSECs activate HSCs by delivering SK1 protein, leading to AKT phosphorylation. SK1 level has been found to be upregulated in both liver tissues and exosomes in CCL4 induced liver injury. Treatment with an SK1 inhibitor has been shown to protect mice from liver fibrosis, indicating another potential therapeutic target [117]. Mechanisms of induction of hepatic fibrosis are presented in Figure 1.

## 8. Role of Apoptotic Bodies in Hepatic Fibrosis

Apoptosis is extremely coordinated and genetically controlled cell death. It is indispensable during the early developmental stage for proper organogenesis. In adults, apoptosis removes the undesirable cells as well as those cells that have been damaged beyond repair. Apoptosis sustains the balance of cells in the human body and is one of the most important aspects in the immune system. A number of discrete morphological changes, including chromatin condensation and marginalization, cell shrinkage, blebbing of plasma membrane, accompanied by DNA fragmentation, membrane alterations and degradation of cellular proteins occur during the process of apoptosis. Finally, the dying cell is fragmented into membrane-bound vesicles comprising reasonably intact organelles and chromatin residues termed as “apoptotic bodies”.

Clearance of apoptotic bodies by phagocytic cells is important as their presence could trigger several pathways leading to inflammation; however, this process may directly stimulate fibrogenesis. During CLD, considerable hepatocyte apoptosis occurs, and phagocytic cells are incompetent in removing the apoptotic bodies leading to accumulation of apoptotic bodies [118]. Afterwards, autolysis of these apoptotic bodies releases their pro-inflammatory contents and induces inflammation. The engulfment of apoptotic bodies is not merely a clean-up process, but it also initiates numerous intracellular signaling cascades in the phagocytizing cell with distinct responses comprising cytokine secretion. Engulfment of apoptotic bodies by KCs enhances expression of death ligand Fas and pro-inflammatory cytokine TNF-α, thus accelerating hepatocyte apoptosis and provoking hepatic inflammation resulting in hepatic fibrosis [119]. Impairment in Fas-mediated apoptosis abridged hepatic fibrosis in a model of experimental extrahepatic cholestasis [120]. Moreover, inhibition of caspases or cathepsin B also reduced fibrosis [121]. Persistent hepatocyte apoptosis owing to hepatocyte specific disruption of Bcl-X_L_ induces liver fibrosis with advanced age, suggesting hepatocyte apoptosis is fibrogenic [122]. Phagocytosis of apoptotic bodies enhances TGF-β production, which is a well-known pro-fibrogenic cytokine and has pro-apoptotic function in the liver. Recently it has been identified that apoptotic bodies induce the release of TGF-β transporting monocytic extracellular vesicles [123]. Engulfment of apoptotic bodies by HSCs induces NADPH oxidase and is associated with liver fibrosis in vivo [124]. Association between apoptosis and fibrosis is bidirectional process, where fibrosis in turn stimulates apoptosis, as shown by the induction of pro-apoptotic gene expression in parenchymal cells after alterations in ECM.

## 9. Role of Inflammasomes in Hepatic Fibrosis

Evidence suggests an essential role of inflammasomes and their downstream effector molecules in liver inflammation and fibrosis [125]. Inflammasomes are intracellular multi-protein complexes that are expressed by innate immune cells, hepatocytes and other non-parenchymal liver cells. They react to cellular danger signals by activating caspase 1, and releasing pro-inflammatory IL-1β and IL-18 cytokines. Out of several inflammasomes, NLRP3 has been increasingly implicated in the pathogenesis of chronic inflammatory liver disease [126]. NLRP3 can induce programmed inflammatory cell death pathways, pyroptosis. In the liver, activated KCs stimulate NLRP3, which results in a wide range of immune responses including production of pro-inflammatory cytokines and chemokines that subsequently induce cell death [127]. NLRP3 initiates inflammation, leading to tissue damage and fibrosis in various conditions including alcoholic steatohepatitis, DILI, and NASH, through IL-17 and TNF-α [128]. NLRP3 mutant mice have been shown to have higher expression of these cytokines that were mainly produced by the infiltrating cells. In mice, persistent activation of NLRP3 inflammasome resulted in severe inflammation, fibrosis and pyroptotic hepatocytes cell death. NLRP3 induced IL-1β production promotes the proliferation and trans-differentiation of HSCs with a drastic increase in fibrogenic markers, comprising metalloproteinases, tissue inhibitor metalloproteinase-1, collagen1α1 and TGF-β, as well as a decrease in BAMBI, a negative regulator of TGF-β signaling [129]. Moreover, NLRP3 activation in the murine model of NASH increases fibrosis, signifying NLRP3 as a potential target to inhibit or reverse the development of fibrotic NASH [130]. Mice that are genetically deficient in any of the components of NLRP3 inflammasome, including caspase or adaptor protein apoptosis associated speck-like protein (ASC), are protected from fibrosis development [131]. Indeed, caspase deficient mice were secured with high fat diet-induced as well as NASH-induced hepatic inflammation and fibrosis [132]. NLRP3 depleted mice are protected from a choline-deficient, L-amino defined (CDAA) diet-induced fibrosis and NASH, while transgenic mice with constitutive global NLRP3 activation develop fibrosis when placed on a short-term CDAA diet [130]. Since global knockout of NLRP3 could not reveal the tissue specific role of the inflammasome, further studies have demonstrated that LX-2 cells (immortalized human stellate cells) and murine HSCs both express the components of NLRP3 inflammasome, and that its activation results in a phenotypic switch from quiescence to myofibroblasts [35]. This activity was abolished in HSCs isolated from ASC-deficient mice, indicating that inflammasome activation in HSC might be sufficient to induce a fibrogenic response. This has been further confirmed in vivo using HSC-specific NLRP3 knock-in mice, in which NLRP3 activation in HSCs increased SMA positive cells and was associated with spontaneous development of fibrosis [34]. Recently, the relationship between NLRP3 inflammasomes/IL-1β pathway with liver damage and fibrosis has been examined in humans. It has been demonstrated that NLRP3 inflammasome components as well as IL-1β levels are increased in the chronic HCV infection and NASH, and are associated with increased markers of fibrosis [133]. Another essential activator of NLRP3 that contributes to CLD are ROS. Elevated levels of NLRP3 and NOX4 (a NADPH oxidase that produce ROS) have been found in α-SMA positive cells [134]. Increased NOX expression and ROS production in HSCs induces NLRP3-mediated HSC activation and increased collagen production. Treatment with antioxidants abrogates NLRP3 activation and intrahepatic collagen synthesis.

## 10. Role of MicroRNAs (miRNAs) in Hepatic Fibrosis

Among the various endogenous factors that regulate gene expression, miRNAs are crucial. miRNA is involved with HSC activation and hepatic fibrosis [135]. During liver damage, several signaling pathways are activated by inflammatory factors that initiate the process of hepatic fibrosis; these signaling pathways are regulated by miRNAs. miRNA are a class of small, endogenous single stranded non-coding RNAs of about 18–25 nucleotides that are involved in numerous biological processes, including cell proliferation, differentiation and apoptosis [136]. miRNA functions in RNA, silencing, as well as providing a post-transcriptional regulation of gene expression. miRNAs are associated with HSC activation and the progression of hepatic fibrosis through binding to signaling molecules. miRNA retains both pro and anti-fibrotic properties, which could be manipulated for molecular targeted therapies of hepatic fibrosis. Various miRNA families, some of which have pro-fibrotic and others anti-fibrotic roles, include: miR-15, miR-21, miR-29, miR-34, miR-199 and miR-200 [137]. The miR-15 family promotes cell proliferation and induces apoptosis, while the miR-29 family regulates the accumulation of ECM and supports apoptosis by modifying PI3-kinase/AKT signaling pathway [138]. Expression of miR-29b has been observed in different hepatic cell compartments of mice, including HSCs, KCs, hepatocytes and LSECs. miR-29 is linked to various signaling pathways such as TGF-β, NF-κB and PI3K/AKT and exacerbates liver fibrosis [139]. The crosstalk between miR-29b and TGF-β/Smad3 signaling occurs in activated HSCs. Smad3 negatively regulates the expression of miR-29b, which directly regulates TGF-β/Smad3 signaling and promotes hepatic fibrosis. A decrease in miR-29b1a in hepatocytes contributes to fibrosis. Genetic knockdown of miR-29b1a increases susceptibility to fibrosis following fibrogenic stimuli. miR-34 activates HSCs and induces fibrosis in rats by targeting acyl-CoA synthetase, which plays an essential role in hepatic lipid metabolism. Furthermore, miR-34 increases the deposition of ECM proteins, as shown in rats. Over expressed miR-34a and miR-34c enhance α-SMA, an important component of hepatic fibrosis, and upregulate the expression of MMPs, including MMP2 and 9 [140,141]. In hepatocytes, miR-34a directly targets caspase 2 and SIRT1, intensifying susceptibility to apoptosis [141]. In addition, miR-21 is involved in HSC activation, collagen synthesis and oxidation [142]. Recently, it has been shown that miR-21 knockout in NASH-associated liver damage results in decreased steatosis, inflammation, and lipoapoptosis, resulting in reduced fibrosis [143]. Moreover, loss of miR-21 expression decreased collagen deposition and fibrotic markers, including TGF-β and α-SMA. There are several other miRNAs with a potential role in hepatic fibrosis, which are summarized in Table 1. Exploring miRNA-based molecular mechanisms may help in the development of new molecular targeted therapy for the treatment of hepatic fibrosis.

## 11. Therapeutic Approaches to Target Liver Fibrosis

Since fibrosis is an outcome of another liver disease, there is no specific treatment for liver fibrosis, and the best possible way to treat or reverse fibrosis is to control the underlying cause. Advances in the understanding of the pathophysiological basis of fibrogenesis are now leading to the development of novel therapeutic approaches. Treatment options for liver fibrosis may include the use of antiviral drugs, alcohol abstinence and weight loss in case of viral hepatitis, alcoholic liver disease and obesity induced fibrosis, respectively [162,163,164,165]. Currently available therapies are mostly directed towards suppressing hepatic inflammation rather than reducing fibrosis. However, substantial recent progress in understanding the pathophysiology of fibrosis has allowed for development of potential anti-fibrotic therapies, including eradication of deleterious stimuli, downregulation of HSC activation and encouragement of matrix degradation [166]. However, in cases of advanced stages of fibrosis, liver transplantation remains the only effective therapeutic option. For the effective treatment of fibrosis, the ultimate drug would be a well-tolerated one with a specific liver target and that promotes the reabsorption of excess interstitial matrix without abolishing the salutary effects of the normal hepatic ECM. However, determining the appropriate targets is critical to develop new therapeutic strategies.

Distinct anti-fibrotic therapies target different areas in the fibrogenic cascade, including collagen synthesis, inhibition of matrix deposition, modulation in HSC activation, matrix degradation, induction of stellate cell death and apoptosis. There are several clinical trials focusing on the development of targeted therapies for different pathological settings that are moving closer to reality, which we have summarized in Table 2 and briefly discussed below.

### 11.1. Targeted Therapies against Nuclear Receptors

HSCs display various nuclear receptors including FXR and peroxisome proliferator-activated receptor γ (PPARγ) [167]. These receptors play a critical role in HSC regulation. The FXR receptor is accountable for maintaining bile acid and cholesterol homeostasis, and regulates the transcription of several genes involved in bile acid synthesis and transport. Activation of FXR in HSCs is associated with a substantial reduction in collagen production [168]. There are several clinical trials targeting FXR receptors in NASH patients (NCT03517540, NCT04065841, NCT02854605, and NCT02548351). Treatment of NASH patients with Obeticholic acid, a FXR agonist, revealed improvement in fibrosis and key components of NASH, and represents a milestone in the development of new therapies for NASH [169]. The FXR receptor also induces the expression of PPARγ. Quiescent HSCs highly express PPARγ, which decreases upon HSC activation. In-vitro treatment with PPARγ agonist restores PPARγ receptor expression, leading to reduced HSC activation and collagen production [170]. Lanifibranor, a PPAR agonist, activates each of the three PPAR isoforms including PPARα, PPARδ, PPARγ and is in clinical trial for the treatment of NASH [171].

### 11.2. Targeted Therapies against HSC Activation

Since HSC activation is considered one of the major pathways to fibrosis, therapeutic approaches are focusing on controlling HSC activation. Understanding the dynamics of HSC activation has assisted in the development of anti-fibrotic therapies. Recently, a review by Nathwani R. et al. elaborated on HSC-targeted anti-fibrotic therapies, which might help in the reversal or abolition of fibrosis. The application of nanoparticle systems has emerged as a quickly evolving area of interest for the safe delivery of many drugs and nucleic acids for treatment of chronic liver disease. Various nanoparticulate systems are primarily focusing on targeting HSCs for the treatment of hepatic fibrosis. Existing nanoparticle therapies are efficient in reversal of early fibrosis and are currently being evaluated in preclinical and clinical trials. Nanoparticulate systems with stimuli sensitive polymers and lysosomes have gained much attention for the treatment of liver fibrosis as well as other diseases[172,173]. Among various nanoparticulate systems, liposomal nanoparticles are in clinical trials and considered a breakthrough in the treatment of hepatic fibrosis [172]. Activated HSCs express surface receptors including integrin α_v_ β_3_ [174], retinol-binding protein receptor [175], PDGF receptor-β [176], CXCR4 [177], CD44 [178] along with other ligands. Conjugated nanoparticles containing anti-fibrotic agents for these receptors may be used to target HSC activation for the treatment of liver fibrosis. CXCR4 antagonist AMD 3100 octahydrochloride hydrate grafted nanoparticles have been shown to inhibit the progression of CCL-4 induced liver fibrosis in mice [179]. Activated HSCs display CD44 that is identified as the main cell surface receptor of hyaluronic acid. Therefore, grafting hyaluronic acid on nanoparticles could be beneficial for the HSC-targeted delivery of anti-fibrotic agents. Recently, hyaluronic acid modified nanoparticles containing curcumin showed efficiency in inducing apoptosis of activated HSCs without affecting quiescent HSCs [180]. Other types of nanoparticles have also been discovered to target HSC by surface engineering with vitamin A. Administration of vitamin A grafted nanoparticles in CCL4-induced liver fibrosis showed significant reduction in collagen production in mice [181]. 

### 11.3. Targeting Therapies against Inflammation and Oxidative Stress

Anti-inflammatory drugs are also a therapeutic strategy for the treatment of liver fibrosis [182,183]. This approach may focus on either directly inhibiting the release of inflammatory cytokines or neutralizing them with receptor antagonists. Increased TNF-α production is one of the primary events in liver injury leading to the release of pro-inflammatory cytokines. Pentoxyfylline, a potent phosphodiesterase inhibitor is effective in inhibiting TNF-α production and suppresses TNF-α mediated inflammatory response [184]. Pentoxifylline is also hepatoprotective by mitigating oxidative stress. Long-term treatment with pentoxifylline has been associated with improvement in histological features and a substantial reduction in liver fibrosis in patients with ALD; therefore, it has been registered in several clinical trials [185]. Finally, other anti-inflammatory agents may also serve as anti-fibrotic therapy. Silymarin that inhibits lipid peroxidation is in clinical trial (NCT03568578) for the treatment of HCV patients.

### 11.4. Targeted Therapies against Renin-Angiotensin System (RAS)

The RAS system is a hormone system that regulates blood pressure and fluid and electrolyte balance, as well as systemic vascular resistance. Upregulation of RAS activity has been observed during hepatic fibrosis [186]. In the liver, the main RAS protein angiotensin II (Ang II) is produced from its precursor angiotensin 1 by the proteolytic cleavage. Ang II performs its function by binding to Ang II type 1 receptor (AT1-R) which is highly activated in activated HSCs [187]. Ang II is responsible for HSC activation, proliferation, contraction and collagen synthesis in both rats and humans [188,189]. Hence, Ang II and its receptor (AT1-R) is critical in mediating liver fibrosis and blockade of AngII and (AT1-R) may be effective in treating liver fibrosis. Candesartan, Ramipril and Losartan, which are angiotensin receptor blockers, are in clinical trials for the treatment of HCV patients. Candesartan inhibits hypertension and has promising results in alcoholic liver fibrosis when used in combination with ursodeoxycholic acid. Losartan showed significant effects in reducing the expression of fibrogenic mediators, decreasing ECM accumulation and inflammation in HCV patients. Furthermore, it decreased oxidative stress in hepatic fibrosis [190]. 

## 12. Conclusions

While chronic liver disease has diverse etiologies, development of hepatic fibrosis is the common outcome. Liver fibrosis has long been considered irreversible; however, studies have demonstrated the potential for reversibility. Current treatment options, particularly those that treat the primary injury, can allow the complete resolution of fibrosis. For instance, antiviral therapies against HBV and HCV, anti-inflammatory therapy with corticosteroids in alcoholic hepatitis, immunosuppressants and corticosteroids in autoimmune hepatitis, ursodeoxycholic acid and FXR agonists in PBC and vitamin E, obesity surgery and lifestyle change in NASH help in the reversal of fibrosis. Among these pathological conditions, the most potent potential for reversibility probably occurs in hepatitis B patients, as shown by a study in which inhibition of HBV allowed the reversion of not only fibrosis but also cirrhosis [191]. In recent times, tremendous progress has been made in understanding the pathophysiology of hepatic fibrosis, which has opened the door for new therapeutic approaches to treat this condition. As HSCs play a key role in the development of fibrosis, advancement in the knowledge of HSCs contribution has provided new targets for anti-fibrotic therapies. Several potential therapeutic targets are emerging in clinical trials. However, continued research is required to identify other mechanisms of liver fibrosis for the development of potential therapies.

## Figures and Tables

**Figure 1 cells-10-01097-f001:**
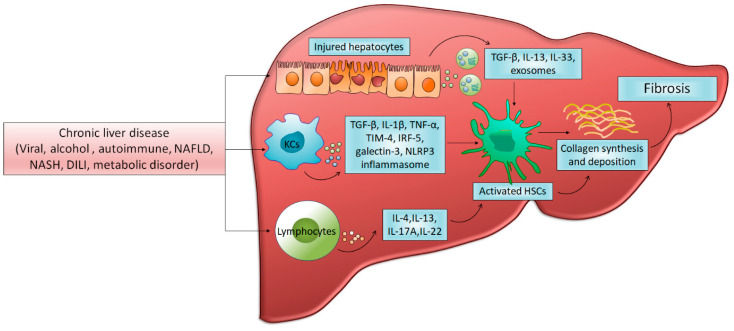
Mechanisms of liver fibrosis. Chronic liver injury mediated by different factors activates several parenchymal and non-parenchymal cells and induce cellular and molecular pathways that encourage hepatic inflammation by producing numerous inflammatory mediators. Extreme inflammation drives hepatic stellate cells activation, which then transform into proliferative and extracellular matrix producing myofibroblast leading to fibrosis and hepatic dysfunction. NAFLD: Non-alcoholic fatty liver disease, NASH: non-alcoholic steatohepatitis, DILI: drug-induced liver injury, KCs: kupffer cells, TGF-β: transforming growth factor beta, TNF-α: tumor necrosis factor alpha, TIM-4: T cell immunoglobulin and mucin-4, IRF-5: interferon regulatory factor-5, NLRP3: NLR family pyrin domain containing 3, HSC: hepatic stellate cell.

**Table 1 cells-10-01097-t001:** Role of miRNAs in hepatic fibrosis.

miRNAs	Role in Hepatic Fibrosis	References
miR-15 family	Cell proliferation, apoptosis, suppression of hepatocyte growth factor, an inhibitor of TGF-β	[136,144]
miR-21	Collagen synthesis and deposition, induction of TGF-β and α-SMA, HSC activation	[142,145]
miR-23a	Activation of PTEN/PI3K/Akt signaling pathway	[146]
miR-29 family	Activation of fibrosis-inducing pathways including TGF-β, NF-κB, PI3K/AKT signaling, induction of ECM related genes, inhibit HSC activation	[138,139]
miR-32	Promote epithelial to mesenchymal transition	[147]
miR-34 family	HSC activation, deposition of ECM proteins, upregulation of MMPs	[140,148]
miR-181	Inhibit Augmenter of liver regeneration, promote epithelial mesenchymal transition, HSC activation	[149,150]
miR-194	Inactivate HSCs, inhibit α-SMA and type 1 collagen	[151,152]
miR-199 and miR-200	ECM deposition, production of pro-fibrotic cytokines	[137,153]
miR-214	HSC activation, ECM accumulation, induction of pro-fibrotic genes	[154,155]
miR-223-3p	HSC activation	[156,157]
miR-378	Induction of NF-κB and TNF-α, inflammation, inhibition of HSC activation	[158,159]
miR-542-3p	Inhibit HSC activation	[160,161]

**Table 2 cells-10-01097-t002:** Clinical trials for different etiology related fibrosis.

Drug	Target	Phase	Trial Number
NASH			
Tropifexor	FXR agonist	II	NCT03517540
Tropifexor	FXR agonist	II	NCT04065841
Cilofexor	FXR agonist	II	NCT02854605
Obeticholic acid	FXR agonist	III	NCT02548351
Cenicriviroc	Antagonist for CCR2 and 5	II	NCT02217475
GR-MD-02	Galectin-3 inhibitor	II	NCT024662967
GR-MD-02	Galectin-3 inhibitor	I	NCT01899859
BMS986036	FGF21 analogs	II	NCT02413372
BMS986036	FGF21 analogs	II	NCT03486912
BMS986036	FGF21 analogs	II	NCT03486899
NGM282	FGF19 analogs	II	NCT02443116
JKB-122	TLR4 antagonist	II	NCT04255069
Lanifibranor	PPAR agonist	III	NCT04849728
GS-4997	Apoptosis signal-regulating kinase	II	NCT02466516
Emricasan	Caspase inhibitor	II	NCT02686762
MGL-3196	Thyroid hormone receptor agonist	III	NCT03900429
CC-90001	Mitogen activated protein kinase-8	II	NCT04048876
Nitazoxanide	Collagen turnover	II	NCT03656068
Selonsertib Firsocostat Cilofexor and combinations	Apoptosis signal-regulating kinase Liver-directed acetyl-CoA carboxylase inhibitor, FXR target	II	NCT03449446
HCV and HCV/HIV			
Candesartan and ramipril	Angiotensin receptor blocker and angiotensin converting enzyme inhibitor	III	NCT03770936
Pirfenidone	Inhibitor of TGF-β	II	NCT02161952
Simtuzumab	LOXL2 antibody	II	NCT01707472
Ursodeoxycholic acid Silymarin, antioxidants and colchicine	Bile duct, Inhibition of lipid peroxidation, oxidative stress, immunomodulatory effect	N/A	NCT03568578
Raltegravir	Integrase inhibitor	II	NCT01231685
Prazosin	Alpha-adrenergic antagonist	II	NCT00148837
Rifaximin	Endotoxin		NCT01603108
Warfarin	Anticoagulation	II	NCT00180674
Losartan	Angiotensin II type 1 (AT1) receptors antagonists	IV	NCT002298714
CHB			
Hydronidone	Inhibitor of TGF-β	II	NCT02499562
Nitazoxanide	Collagen turnover	II	NCT03905655
ALD			
Profermin	Dysbiotic microbiota	N/A	NCT03863730
Ciprofloxacin	Bacterial DNA topoisomerase and DNA-gyrase	I	NCT02326103

N/A, not applicable; FXR, Farnesoid X receptor; FGF, Fibroblasts growth factor; LOXL2, Lysyl oxidase-like-2; PPAR, peroxisome proliferator-activated receptor.

## Data Availability

Not applicable.

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
