# Peer review of "Pathophysiology and Treatment Options for Hepatic Fibrosis: Can It Be Completely Cured?"

_cells, 2021, doi:10.3390/cells10051097_

Round 1
Reviewer 1 Report
The present review titled Pathophysiology and Treatment Options for Hepatic Fibrosis: Can it be Completely Cured? The key cores are Pathophysiology of Hepatic Fibrosis and Treatment Options for Hepatic Fibrosis.
For the parts of Pathophysiology of Hepatic Fibrosis, the author only listed
Role of Liver-Resident Macrophages, Kupffer Cells (KCs), in Hepatic Fibrosis
Role of Liver-Resident Lymphocytes in Hepatic Fibrosis
The injured hepatocytes undergo apoptosis, while sinusoidal endothelial cells experience the loss of fenestrae, known as capillarization of the sinusoids.
I strong suggest the author list Role of hepatic stellate cells in Hepatic Fibrosis
And Role of liver sinusoidal endothelial cells in Hepatic Fibrosis
The HSC is the central effector in hepatic fibrosis and undergoes activation through a two-phase process. Initial liver injury results in hepatocyte cell apoptosis with generation of apoptotic bodies, reactive oxygen species, and paracrine stimulation of HSCs. Additionally, LPS from the gut can simulate HSCs.
Reference as below:
Liver fibrosis and hepatic stellate cells: Etiology, pathological hallmarks and therapeutic targets
World J Gastroenterol. 2016 Dec 28; 22(48): 10512–10522.
Targeting Hepatic Stellate Cells for the Treatment of Liver Fibrosis by Natural Products: Is It the Dawning of a New Era?
Front. Pharmacol., 30 April 2020 | https://doi.org/10.3389/fphar.2020.00548
LSEC are also main players in fibrosis resolution. They maintain liver homeostasis and keep hepatic stellate cell and Kupffer cell quiescence. After sustained hepatic injury, they lose their phenotype and protective properties, promoting angiogenesis and vasoconstriction and contributing to inflammation and fibrosis.
Reference as below:
Hepatology. 2015 May; 61(5): 1740–1746.
Published online 2015 Mar 23. doi: 10.1002/hep.27376
PMID: 25131509
Liver Sinusoidal Endothelial Cells in Hepatic Fibrosis
Three dimensional visualization of arsenic stimulated mouse liver sinusoidal by FIB- SBFSEM approach. Protein &Cell. (2016).
Quantitative proteomic study of arsenic-treated mouse liver sinusoidal endothelial cells using a reverse super SILAC method. Biochemical and Biophysical Research Communications. 2019 Jun 25;514(2):475-481.
A New Insight into the Impact of Different Proteases on SILAC Quantitative Proteome of the Mouse Liver. Proteomics. 2013 Aug; 13(15). (#Co-first author).
Does Mechanocrine Signaling by Liver Sinusoidal Endothelial Cells Offer New Opportunities for the Development of Anti-fibrotics?
Front. Med., 09 January 2020 | https://doi.org/10.3389/fmed.2019.00312
Only those major concerns addressed , then we could reconsider to accept this review.
Author Response
We sincerely appreciate the reviewer’s comment. As per the reviewer’s suggestion, we have included the roles of hepatic stellate cells and liver sinusoidal endothelial cells in hepatic fibrosis and revised the manuscript accordingly. We have included hepatic stellate cells on page 4 (lines 155-213) and liver sinusoidal endothelial cells on page 5 (lines 214-250) and included the references. All the changes are highlighted in red.
Reviewer 2 Report
The review manuscript entitled “Pathophysiology and Treatment Options for Liver Fibrosis: It Can Be Completely Cured” by Khanam et al. treats the mechanisms for the development of liver fibrosis regardless of the disease responsible for this fibrosis.
MAJOR COMMENTS:
-For this reviewer, approaching fibrosis as an entity is initially incorrect, as currently, each disease has its particular mechanisms responsible for the onset of fibrosis and, therefore, from a medical, or therapeutic point of view, it is necessary to focus on these mechanisms responsible for fibrogenesis, or lack of ECM removal, to be able to improve not only fibrosis, but the complications that can arise in each particular disease. Fibrosis derived from hepatitis C virus, is not the same as fibrosis derived from NAFLD or induced by drug-induced liver disease. It would be interesting to review strategies to eliminate the resilient fibrosis that remains after an already cured disease, but seeing the absence of therapeutic options offered by the authors, this does not seem to be the case.
-Irrespectively of the comment above, the review in itself has several flaws. The introduction is not bad, but the different subsections (Kuppfer Cells, Lymphocytes, Inflammasomes, miRNAs, and Exosomes) just deal with giving and enormous amount of information, sometimes incoherent and with many errors in interpretation, that is difficult to digest, probably due to the fact that since fibrosis is treated as an entity the authors are mixing aspects that are relevant for HCV with others that are specific for NAFLD…and this, makes the review difficult to read. Regarding the section about treatment options, the authors just recommend the use of nanoparticles for the safe delivery of drugs (which ones?), and anti-inflammatory drugs (for example pentoxyfilline), and that is all. The truth is that given the nature of the previous sections, describing the “players” involved in the fibrotic process, I would have expected that the authors propose more therapeutic options.
-Finally, in the title the authors ask if Fibrosis: Can it be completely Cured?, a provocative question, that after reading the review remains unanswered and has not been properly addressed, just a mere phrase in the conclusions sections “Liver fibrosis has long been considered irreversible; however, studies have demonstrated the potential for reversibility”
MINOR COMMENTS:
Line 45, and line 79: Kupffer with capital K
Line 54: Add + after CD4 or CD8
Line 69: “…through TNF signaling”, I believe this is not correct
Line 117: Please define Margatoxin
Line 153-156: Please remove the phrase “This section may be divided by subheadings. It should provide a concise and precise description of the experimental results, their interpretation, as well as the experimental conclusions that can be drawn.
Line 177 and line 285: Please remove ()
Line 316: Please define CLD
Author Response
Answers to the major comments
- We appreciate the reviewers' comment. As per the reviewer’s suggestion, we included different pathological mechanisms associated with hepatic fibrosis including viral infections, alcoholic liver disease, NAFLD/NASH and autoimmune hepatitis and described them under the heading “Triggers of Hepatic fibrosis” (Pages 2-4, lines 80-155) All the changes are highlighted in red. Since in this review we discussed other aspects of hepatic fibrosis including phenotypic and functional alteration in parenchymal as well as non-parenchymal cells and its role in hepatic fibrosis, therapeutic approaches required to treat hepatic fibrosis, we briefly discussed about the underlying cause of hepatic fibrosis. As per the reviewer’s suggestion, we elaborated on different nanoparticles that can be used for the treatment of hepatic fibrosis targeting different surface receptors on HSCs.
- We are thankful to the reviewer for the comment. As per the reviewer’s concern, we revised the section about treatment options and comprehensively described different target based therapeutic strategies to treat liver fibrosis under the headings, (1) Targeted therapies against nuclear receptors, (2) Targeted therapies against HSC activation, (3) Targeted therapies against inflammation and oxidative stress, (4) Targeted therapies against renin-angiotensin system (pages 13-15). We have also included a table (table 2) where we mentioned different drugs, which are in clinical trials, for the treatment of different etiology related liver fibrosis (pages 15-17).
Moreover, we revised the conclusion section in an attempt to answer the question “Can it be completely cured?” We highlighted all the changes in red.
Answers to the minor comments
1. As per your kind suggestion, we replaced Kupffer cells with capital K
2. We have added added + after CD4 and CD8.
3. As per your kind suggestion, we double checked the literature and revised the sentence accordingly.
4. We have defined Margatoxin.
5. As per your kind suggestion, we have removed the phrase.
6. We have removed the () on both the places.
7. We have defined CLD.
Reviewer 3 Report
This review is very well written and covers important and interesting aspects of liver fibrosis development. In general, I do not have problems with this manuscript. Taking into account the multifaceted nature of liver fibrosis, we understand that it is difficult to cover its pathogenesis in full in the frame of one review. However, certain things need to be done for clearer presentation purposes:
- The triggers of liver fibrosis, such as alcohol, NASH, viruses (HCV, HBV, HIV), etc. should be summarized to make clear how they contribute to fibrosis propagation.
- In addition to exosomes, apoptotic bodies (or apoptotic hepatocytes) play a significant role in HSC activation and liver fibrosis development. It will be nice to expand on that. There are some papers in the settings of HCV and HIV infections.
- The title of the manuscript is: “Pathophysiology and treatment options for hepatic fibrosis: can it be completely cured?” While it sounds attractive, I do not think that the manuscript answers the question whether hepatic fibrosis can be completely cured. Also, the Therapeutic Approaches part should be expanded to make clear which approaches are at a pre-clinical stage and which demonstrated their clinical efficiency. By the way, Obeticholic acid used in NASH patients and in vitro for HIV and alcohol as an anti-fibrotic drug and is about to be approved by FDA has not been even not mentioned.
All suggested changes are minor and do not affect overall excellent impression from this manuscript.
Author Response
-
We thank the reviewer for the comment. As per the reviewer kind suggestion, we have included the triggers for hepatic fibrosis such as HBV, HCV, alcohol, NAFLD/NASH and autoimmune hepatitis. We described them under the heading, “Triggers of Hepatic Fibrosis” on pages 2-4 (lines 80-155) and highlighted it with red color.
-
As per the reviewer’s kind suggestion, we also included the role of apoptotic bodies on pages 10-11 (lines 442-474). Changes are highlighted it red color.
-
We are thankful to the reviewer for the comment. As per the reviewer’s concern, we revised the therapeutic approaches section and comprehensively described different target based therapeutic strategies to treat liver fibrosis under the headings, (1) Targeted therapies against nuclear receptors, (2) Targeted therapies against HSC activation, (3) Targeted therapies against inflammation and oxidative stress, (4) Targeted therapies against renin-angiotensin system (pages 13-15). We have also included a table (Table 2) where we mentioned different drugs, which are in clinical trials, for the treatment of different etiology related liver fibrosis (pages 15-17). As per the reviewer’s suggestion, we discussed about Obeticholic acid as a therapeutic option for the treatment of NASH and included it in the text as well as table 2. Moreover, we revised the conclusion section in an attempt to answer the question “Can it be completely cured?” We highlighted all the changes in red color.
Round 2
Reviewer 1 Report
The author add the section of Role of LSECs in Hepatic Fibrosis, however , I suggest the author update all LSEC related publications and cite of them, such as.
Three dimensional visualization of arsenic stimulated mouse liver sinusoidal by FIB- SBFSEM approach.
Quantitative proteomic study of arsenic-treated mouse liver sinusoidal endothelial cells using a reverse super SILAC method.
I am happy to reconsider it after the author revision again.
Reviewer 2 Report
The authors have extensively amended the manuscript according to the reviewers' comments and it has improved notably.